# Exploring the Visual Guidance of Motor Imagery in Sustainable Brain–Computer Interfaces

Cheng Yang [1], Lei Kong [2,†], Zhichao Zhang [2,†], Ye Tao [1] and Xiaoyu Chen [1,*]

1   Department of Industrial Design, Zhejiang University City College, Hangzhou 310015, China
2   Department of Industrial Design, College of Computer Science and Technology, Zhejiang University, Hangzhou 310027, China
*   Correspondence: rainchard@zju.edu.cn
†   These authors contributed equally to this work.

**Abstract:** Motor imagery brain–computer interface (MI-BCI) systems hold the possibility of restoring motor function and also offer the possibility of sustainable autonomous living for individuals with various motor and sensory impairments. When utilizing the MI-BCI, the user's performance impacts the system's overall accuracy, and concentrating on the user's mental load enables a better evaluation of the system's overall performance. The impacts of various levels of abstraction on visual guidance of mental training in motor imagery (MI) may be comprehended. We proposed hypotheses about the effects of visually guided abstraction on brain activity, mental load, and MI-BCI performance, then used the event-related desynchronization (ERD) value to measure the user's brain activity, extracted the brain power spectral density (PSD) to measure the brain load, and finally classified the left- and right-handed MI through a support vector machine (SVM) classifier. The results showed that visual guidance with a low level of abstraction could help users to achieve the highest brain activity and the lowest mental load, and the highest accuracy rate of MI classification was 97.14%. The findings imply that to improve brain–computer interaction and enable those less capable to regain their mobility, visual guidance with a low level of abstraction should be employed when training brain–computer interface users. We anticipate that the results of this study will have considerable implications for human-computer interaction research in BCI.

**Keywords:** sustainable living; EEG; motor imagery; visual guidance; mental load; ERD

## 1. Introduction

People who are severely physically disabled, those who have had strokes, older people with limited mobility, and other groups face significant obstacles or limitations when it comes to taking care of themselves, moving around, and participating in social activities, necessitating the assistance of others and the proper social security institutions [1]. Brain–computer interface (BCI) technology uses EEG signals to achieve communication between the human brain and computers or other electronic devices and can effectively help the people mentioned above to interact with the outside world [2]. With the help of BCIs, people with a range of motor and sensory impairments are able to interact, communicate, and operate mobility aids in their surroundings [3]. Additionally, BCIs can aid in fostering the independent living of the disabled, reduce the personnel costs of relevant social security agencies, and free up resources that can be applied to other facets of urban life [4]. Recent studies have shown that EEG signals of motor imagery (MI) can be used as control sources in the construction of BCIs, which holds promise for the recovery of motor function [5,6]. As a standard paradigm in brain–computer interfaces [7], MI has rapidly developed in recent years. Underlying this rapid development is the ability of MI to trigger contralateral explicit event-related desynchronization (ERD) and, in some cases, simultaneous ipsilateral event-related synchronization (ERS) by unilateral imaging movements [8] For instance,

when picturing unilateral hand movements, the energy of mu rhythms (8–12 Hz) and beta rhythms (14–30 Hz) in the contralateral brain region is decreased (ERD), whereas the energy of mu rhythms and beta rhythms in the ipsilateral motor–sensory areas is increased (ERS) [9]. The spontaneity and classifiability of MI make it a critical factor in ensuring the availability and smoothness (the efficiency of information transmission) of the machine subsystem in BCI systems. Much of the current research on motion imagery has focused on the separability of MI, enhancing accuracy by examining feature extraction [10], channel selection [11], and classification methods [12].

Instead, it is typically essential to consider the three subsystems of person, machine, and environment, which are in a connection of reciprocal influence and interaction. It is necessary to evaluate the three subsystems because the motor imagery brain–computer interface (MI-BCI) cannot be studied as a system for human–machine interaction by focusing only on the machine subsystem [13]. The system's performance depends on each subsystem functioning consistently and on how well they communicate with one another. Human factors professionals suggest that when a system involves the behavior of a user, that user is one of the most critical subsystems that must be taken into account when assessing system performance [14]. To understand and forecast user performance, it is frequently helpful to measure, estimate, or analyze the user's mental load [15] because user performance can be highly variable and unreliable. The secret to efficient system design is understanding how the user's mental load affects operator performance and, by extension, system performance [16].

MI is a complex mental task [17]. The environment, user's mental state, number of tasks, task difficulty, etc., may influence the user's psychology, affecting the user's performance and, ultimately, the system's accuracy [18]. Usually, MI-BCI employs visual guidance to accurately direct the user through the task and lessen its difficulty. Therefore, before the MI-BCI can be utilized, there must be a period of user training [19]. User training will make it easier for the user to become proficient at performing BCI and MI tasks independently. However, there has been little research to investigate how the user's mental load affects the system's overall performance or how the user subsystem of the BCI system and MI tasks relates to the user. Therefore, by studying the psychology of the user and the mechanisms that control the BCI system, we intend to analyze the impact of visual guidance on the user's mental load in MI tasks and create a successful system. We also hope to assist users in living sustainably autonomous lives using BCIs, and we plan to take the abstraction level of visual guidance as the breakthrough point. The abstraction level of visual guidance refers to the generalization level of different expression models in the cognitive process of visual guided motor imagery tasks.

In summary, the contributions of this paper are as follows:

1.  We demonstrate that the brain activity and mental load during MI have significant differences among the three levels of abstraction of visual guidance. Our results suggest that suitable visual guidance would help users to increase brain activity and reduce mental load during MI.
2.  We provide evidence that a low level of abstraction of visual guidance influences the classification accuracy of MI-BCI compared with the high-abstraction paradigm. Our findings suggest that suitable visual guidance would help users to achieve better classification performance on MI-BCI.
3.  We propose that brain activity and mental load correlate with the classification accuracy of MI-BCI. Our results suggest that suitable visual guidance would help both the user and the machine for sustainable system work.

This paper is organized as follows. Section 2 summarizes existing studies related to MI-BCI and mental load based on electroencephalographs (EEG). In Section 3, we define our research hypotheses and research objectives, and we develop multiple indicators and methods of brain activity, mental load, and classification performance. Section 4 presents the ERD values that represent brain activity, the PSD values that represent mental load, and the accuracy results classified by SVM. Section 5 describes several correlation analyses

between brain activity, mental load, classification performance, and discussions. Finally, the paper concludes in Section 6.

## 2. Related Work

### 2.1. Related Work of MI-BCI

MI-BCI requires thorough training before the operation, for both the user and the computer [19,20] [NO_PRINTED_FORM]. On the one hand, to correctly transmit recognizable patterns of brain activity to the computer, the user must learn the MI-BCI manipulation technique [21]. On the other hand, the computer has to be able to recognize the variations in brain activity that take place as the user engages in various mental tasks [22].

Some research projects aimed at enhancing the performance of MI-BCI concentrate on the feature extraction of EEG data to achieve enhanced classification accuracy by choosing particular times or particular channels. An algorithm was proposed by Liang et al. [23] to automatically extract the best frequency and temporal bands unique to the subject to distinguish between the ERD patterns produced by left- and right-handed MI. This study examined how left- and right-handed MI's sensorimotor EEG rhythms were affected by employing object-oriented motion as a visual cue in a virtual environment. The results indicated the increased reliability and efficiency of MI-based BCI systems. Most researchers, including Tang et al. [24], have concentrated more on the advancement of categorization algorithms. For instance, a five-layer convolutional neural network (CNN) was designed explicitly for motor imagery classification based on EEG signals' combined temporal and spatial characteristics. After the technique was applied to experimental and public datasets, a classification model was created. The work offers a theoretical foundation and technical justification for using brain–machine interface technology in the exoskeleton sector of rehabilitation. Cheng et al. [25] created a convolutional neural network with three convolutional layers, three pooling layers, and two fully connected layers to classify the EEG signals in motor imagery under the assumption that fewer EEG acquisition channels would result in more classifications. Experiments were designed and carried out for four states of imagining: the left hand, right hand, and foot in motion and resting state. With an average classification recognition rate of 82.81%, a convolutional neural network-based classification model was built using the pertinent EEG data that had been collected.

A few studies have also been carried out to enhance the general usability of BCI systems by concentrating on the user control mechanisms of BCI [21]. According to several studies, the human brain has a mirror neuron system where visually guided MI can modify sensorimotor EEG rhythms [26]. In numerous studies on BCI system users, visual guidance has been used to accurately guide subjects during motor imagery. These studies prove that visual guidance to aid users during motor imagery can help users operate a BCI system with significantly higher operability. For instance, Liang et al. [27] trained and classified MI tasks in a 3D virtual environment utilizing non-object-oriented, static, and dynamic object-oriented scenarios. They did this using single- and multi-subject BCI paradigms. It was shown that object-oriented scenarios, whether static or dynamic, can offer better classification accuracy than non-object-oriented scenarios. Both can speed up reaction time and be used with sparse training data. In Sun et al.'s [28] study, ten patients with post-stroke upper limb motor dysfunction were split into two groups and given MI exercises and regular rehabilitation routines for four weeks. The experimental group was requested to complete the same MI under synchronous visual guidance, while the control group was required to complete the MI under asynchronous visual assistance. The findings demonstrated that the ERD pattern in the experimental group not only had a bigger amplitude and longer duration but also contained more frequency components.

### 2.2. Related Work of Mental Load

The term "mental load" describes the cognitive effort required to complete a task over a specific amount of time [29]. The current study aims to understand how to control an MI-BCI system by determining how many cognitive operations are needed to perform

left- and right-hand MI tasks. The mental load is the workload a user experiences while executing a task, and it is affected by the user's surroundings and outside influences [30]. The number of tasks, their level of complexity, and external factors like the temperature and lighting all impact mental load. Personal aptitude (both physical and mental), training, experience, fatigue, stress, and personality are internal variables that influence mental load [31,32]. An excessive or insufficient mental load can cause human error, according to research on mental load in specific work scenarios [30]. Working memory and attention are two cognitive processes involved in the control of BCIs [33]. Due to this, task performance efficiency and duration can be significantly increased by developing MI tasks with the proper intensity of mental load needs. It is necessary to measure the user's mental load for various visually guided MI tasks to guide the user through the MI task using the visual guidance that causes the lowest mental load. Using visual guidance that causes low mental load can avoid the need to obtain high-quality EEG signals except when the subject is over-motorized.

Cognitive load, often known as mental workload, has frequently been examined and assessed using EEG [34]. In several scientific domains, such as sustainable work [35], driving status detection [36], visualizing impacts [37], and treating cognitive impairment [38], the direct use of EEG to quantify mental load has not only been demonstrated to be feasible but also found practical applications. According to research on burnout in sustainability and sustainability psychology, López-Nez et al. [35] focused on the prevention of burnout and the promotion of individual and organizational well-being for sustainable development. They examined the relationship between job demands (workload), human resources (psychological capital), and burnout. Kim et al. [39] examined EEG data gathered during driving tests on city roads. They considered the variation in EEG values across drivers and used the EEG rate of change to measure cognitive load. A reference interval was chosen for each of the five extracted behavioral sections: a left-turn section, a right-turn section, a rapid acceleration section, a rapid deceleration section, and a road change section, and the EEG values of the extracted sections were compared with the EEG values of the reference interval to determine the EEG rate of change, which was then statistically analyzed. To inform safe driving while considering driving workload, the study's results are being utilized to understand the cognitive aspects of driver behavior-induced driving workload in vehicle information systems. By applying the deep learning of EEG data to detect cognitive driver load under high- and low-load tasks, Almogbel et al. [33] created a method for detecting cognitive driver load. They gathered their data during various driving tasks on a high-fidelity driving simulator. According to preliminary studies using data from only four EEG channels, the system can accurately recognize the driver's cognitive load, and there is much room for improvement.

Furthermore, Rusnock et al. [29] suggested a method of estimating workload continuously without interrupting the operator, making this continuous workload assessment a workload profile with a focus on five areas that cannot be completed using existing workload measurements. Additional studies use EEG to quantify mental load and apply it practically. For instance, Zammouri et al. [40] investigated the potential use of data collected noninvasively from the human cortex to develop a BCI capable of estimating brain workload and mental effort during cognitive tasks and ultimately used this EEG-based workload classifier to evaluate re-education therapies used in a physiotherapy center for children with cognitive impairment.

## 3. Experiments

### 3.1. Research Objectives

In this study, EEG was utilized to examine the impact of the level of abstraction of visual guidance on the user's mental load and served as the experimental data for the motor imagery task. The goal of this study was to determine the following: (1) whether the level of abstraction of visual guidance affects subjects' brain activity during motor imagery; (2) whether the level of abstraction of visual guidance affects the classifiability of motor

imagery and consequently MI-BCI performance; (3) whether the level of abstraction of visual guidance affects the variability in subjects' mental load; and (4) whether there is a correlation between participants' mental load and the classifiability of their motor imagery.

### 3.2. Research Hypothesis

Research in related domains has established direct factors (motivation, attentiveness, etc.) that affect how users learn and carry out tasks. In research on mental load, tasks requiring significant cognition such as attention span tend to put more mental strain on people. According to neuroscience, the prefrontal lobe of the human brain, which is active during cognitive processes, is closely associated with cognition. Additionally, tasks involving the visual domain draw and hold participants' attention, and drawing attention increases mental load. The impact of visual guiding in MI-BCI on experimental outcomes, such as performance and user mental load, has not yet been examined in any studies. Consequently, this essay puts forth the following presumptions:

**H1.** *Brain Activity: During the motor imagining task, visual guidance at various degrees of abstraction impacted the subject's brain activity.*

**H2.** *Mental Load: During the motor imagery task, respondents' mental loads were affected by visual guidance at various degrees of abstraction.*

**H3.** *MI-BCI Performance: On the motor imagery task, MI-BCI performance was influenced by visual guidance at various degrees of abstraction.*

**H4.** *Brain Activity, Mental Load, and MI-BCI Performance: In visually guided motor imagery tasks with varying degrees of abstraction, there is a correlation between the performance of MI-BCI, the brain activity of the subjects, and the mental load.*

### 3.3. Subjects and Data Acquisition

There were 17 total subjects, 11 men and 6 women, who ranged in age from 18 to 26 and had an average age of 21.65 years (SD = 2.42). All participants were right-handed and had normal or corrected eyesight, normal color vision, and no history of mental illness or psychosis. They were instructed to abstain from getting a perm or coloring their hair in the three months before the experiment and from consuming stimulating drinks like wine or coffee the day before.

Before starting the EEG tests, the subjects were aware that the tests would not harm them psychologically. The process and specifics of the experiment were explained to the subjects, who agreed to the experiment before they read and signed the informed permission form. The test participants were instructed to execute a MI mental task while sitting in a chair 50 cm from a computer screen, facing the screen, in a quiet environment.

### 3.4. Experimental Procedure

The experiments used the classical Graz paradigm of asynchronous MI experiments [41]; BCI research has frequently employed this paradigm. To lessen the impact of visual residuals on MI and to better match the scenario to the actual application, the paradigm mandates that subjects perform the MI task after watching the visual guidance and that they do not perform visual guidance while performing the MI task. In the research on brain science and semeiology, specific brain regions such as the hippocampus change while watching symbols and stimuli with different levels of abstraction [42]. In psychological research, different moving stimuli have different abstract memory representations [43]. Since our MI tasks were under the guidance of visual stimuli, we took the abstraction level of visual guidance as the breakthrough point. During the experiment, participants were required to complete three different types of visual guidance MI tasks at various levels of abstraction: (1) high level of abstraction, which employed an abstract guide to cue left-

and right-hand gripping based on the motion of a ball; (2) medium level of abstraction: a more abstract guide that used line drawings of hand grasp animations to cue left- and right-hand grasping; (3) low level of abstraction: a nonabstract (figurative) task that cued the left and right hands using real-world hand grip animations. The three types of cues are shown in Figure 1, where the cues in the medium and low abstraction groups are shown in the first view. In the experiment, participants were instructed to maintain a relaxed state throughout, view one of the three cues, and then visualize the left and right hands holding an object on a blank screen after the signals had vanished. The cues were animated and occurred randomly.

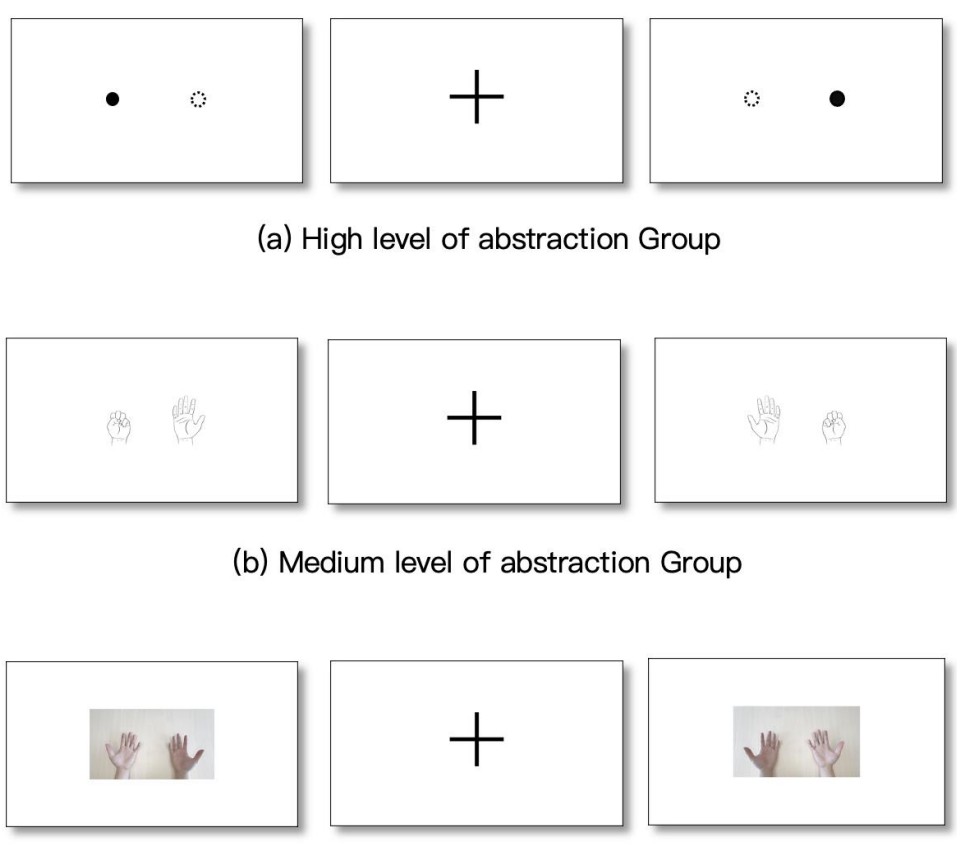

(a) High level of abstraction Group

(b) Medium level of abstraction Group

(c) Low level of abstraction Group

**Figure 1.** Three levels of abstraction of visual guides. (**a**) High level of abstraction, which employs an abstract guide to cue left- and right-hand gripping based on the motion of the ball; (**b**) medium level of abstraction: a more abstract guide that uses line drawings of hand grasp animations to cue left- and right-hand grasping; (**c**) low level of abstraction: a non-abstract (figurative) task that cues the left and right hands using real-world hand grip animations.

Figure 2 shows the timing diagram for this experimental paradigm. A single-trial experiment took 8 s to complete and had 4 phases: a 1 s preparation phase, a 2 s viewing guidance phase, a 3 s motor imagery phase, and a 2 s rest phase. During the preparation stage, a cross was placed in the center of the screen to draw the subject's attention and hold it there. An animation was played during the viewing guidance phase to show the subject their left and right direction. The subject watched the animation and marked the direction. The subject then imagined the action of the proper hand grip during the motor imagery phase, with the screen blank to avoid visual residue. The screen was blank, and only the word "relax" was displayed in the center after the visuals. The left- and right-hand grip MI appeared randomly repeated 144 times, 72 times for the left-hand grip imagery and 72 times for the right-hand grip imagery. The 144 trials were divided into 4 runs, with 36 trials in each run. There was a brief pause between each run determined by the

subject that did not last longer than 2 min. We conducted a motor imagery pre-experiment before the formal experiment. The subjects executed ten motor imagery exercises with their right and left hands before the formal experiment start. After the pre-experiment, we interviewed subjects to understand the vividness of their motor imagery. If subjects experienced difficulties at this stage, we terminated the experiment. Considering the number of exercises and the time required, the entire experiment lasted roughly 30 min.

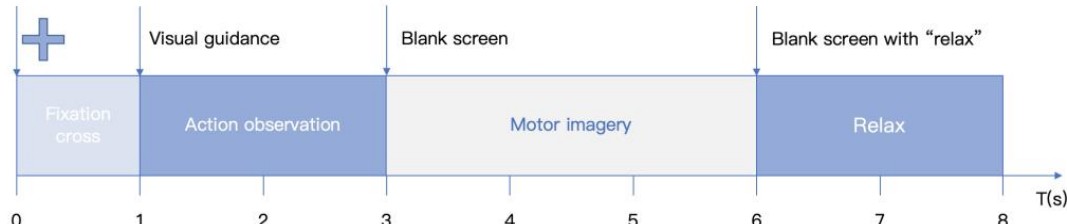

**Figure 2.** Experimental timing diagram. A single-trial experiment took 8 s to complete and had 4 phases: a 1 s preparation phase, a 2 s viewing guidance phase, a 3 s motor imagery phase, and a 2 s rest phase.

The subjects had to complete 3 tests totaling 432 trials over around 90 min with 3 visual guidance events in each experiment. To lessen the impact of fatigue on the outcomes, the three trials were carried out on three different days. The three trials were arranged in a Latin square design to lessen the impact of the experiments' sequence on the outcomes. We ultimately collected 7344 trials' worth of data from 17 participants.

### 3.5. EEG Signal Recording and Data Pre-Processing

The EEG signals were recorded using BrianVision Recorder 2.1 software (Brain Products GmbH, Munich, Germany), and the experiments were carried out using BP actiCHamp amplifiers. BP actiCAP standard 32-channel EEG caps were used to record data from the following electrodes: FP1, FP2, F7, F3, Fz, F4, F8, FT9, FC5, FC1, FC2, FC6, FT10, T7, C3, C4, T8, TP9, CP5, CP1, CP2, CP10, P7, P3, Pz, P4, P8, O1, Oz, and O2. The experiment's online recording used Fz as the reference electrode, and the electrodes were placed in the experiment's standard location for the global 10/20 system. The ground electrode is between the two electrodes, Fp1 and Fp2, in the middle of the forehead and three fingers above the center of the eyebrow. The sampling frequency was 500 Hz, and the impedance was adjusted to below ten kΩ. E-prime 3.0 was used to implement the experimental stimulus presentation. EEG data pre-processing was done using BrianVision Analyzer 2.1 software (Brain Products GmbH, Munich, Germany).

All experimental data were preprocessed offline. The first step was to re-reference the data from the Fz channel and use the two electrodes, TP9 and TP10, as the new reference electrodes since the new electrodes contained less brain activity and were farther from the brain regions we were interested in [44]; the second step was to perform band-pass filtering from 0.5 to 40 Hz and remove signals above 40 Hz. The filter used here was the finite impulse response (FIR) filter based on the hamming window; the third step was to remove the artifacts, and we performed an independent component analysis (ICA) on the raw EEG signals; the fourth step was to segment the EEG signals into epochs, where the point where the visual guidance appears is used as the 0 points and the time from 1 s before to 5 s after it is taken as an epoch; the fifth step was baseline correction, that is, setting the baseline to be 500 ms after the cross appears and then correcting each epoch segment within that time frame.

### 3.6. Measurements

#### 3.6.1. ERD Measurements

The μ-rhythm (8–12 Hz) is a crucial information-processing mechanism that converts perception into action and is intimately connected to MI activity. It represents the down-

stream modulation of mirror neuron activity through primary sensorimotor regions [45]. The degree of neuronal excitation in cortical areas can be determined directly from the absolute value of the μ-rhythm ERD characteristic [28]. As a result, the motor imagery EEG's μ-rhythm ERD value was selected as a gauge of brain activity.

The ERD of EEG is defined as the absolute ratio of the reduction in rhythmic energy between the occurrence interval of motor imagery and the resting-state interval (reference interval) of equal length before the imagery begins:

$$E = |W| = \frac{1}{a-b} \left| \sum_{b}^{a} (A_j - R) / R \right| \tag{1}$$

In the formula $A = \frac{1}{N-1} \sum_{i=1}^{N} \left( x_{ij} - \bar{x}_j \right)^2$, $R = \frac{1}{k} \sum_{r_0}^{r_0+k} A_j$, where $N$ is the number of single experiments, $[a, b]$ is the motor imagery interval, $x_{ij}$ is the filtered band-pass value of the data obtained by sampling the $j$-th experiment for the $i$-th experiment , $\bar{x}_j$ is the average value of $x_{ij}$ obtained from the $j$-th sampling of all experiments, $R$ is average energy value during resting period $[r_0, r_{0+k}]$, $W$ is the energy reduction rate.

3.6.2. Mental Load Measurement

One of the ergonomics concepts that has been the most studied and utilized is mental load, yet there is no universally agreed-upon description of it. In general, mental load can be defined as the quantity of cognitive resources used during a specific task [29].

Overloading or underloading the mind can have various negative implications in real-world production and living [30]. It is crucial to effectively gauge mental load and create manageably challenging assignments. Currently, there are three basic ways to quantify mental load: using a subjective scale, performance metrics, and physiological signal analysis [36,46,47]

Every method of measuring mental load has benefits and drawbacks. However, this research's assessment method is based on physiological signals: EEG. The multiple representational dimensions of mental load can be connected with the neurophysiological characteristics acquired from the EEG signal. EEG-based mental load measures gather neurophysiological signals from various brain regions and typically employ two EEG features. One is in the time domain, like waveforms and amplitudes of event-related units (ERPs), and is sensitive to the amount of cognitive stress over time [48]. The other is frequency domain data, which focus more on the theta (4–8 Hz) and alpha (8–13 Hz) bands to calculate the total energy levels in those bands over time [49]. Power spectral density analysis was used to analyze the EEG data in this study, which placed more emphasis on the frequency domain data.

The PSD of the EEG signal is created by analyzing the frequency domain components of the signal using power spectra. It may be used to compare PSDs produced under various levels of abstraction of visual guidance to assess the mental load. There are two approaches for determining PSD: parametric and nonparametric estimation. Welch power spectrum estimation, a nonparametric estimate technique based on the periodogram, was selected to determine the PSD of the EEG signal in this work.

To represent the distribution of signal energy (density) and phase information in the time-frequency domain, a joint time-frequency distribution function was first developed. This time-frequency distribution can determine the signal power spectrum density and other properties corresponding to a specific time and frequency. The calculation formula is shown as Formula (2):

$$TF_{\text{mean}} (f, t) = \frac{1}{n} \sum_{m=1}^{n} \left( F_m(f, t)^2 \right) \tag{2}$$

where $n$ represents the total number of subjects and $F_m(f, t)$ represents the power spectrum of the $m$-th subject at frequency $f$ and time $t$. The power spectrum can be calculated using the short-time Fourier transform (STFT). The STFT works by breaking a long-term signal

into numerous equal-length shorter signals. Each shorter signal is then given a separate Fourier transform calculation. The calculation formula is shown as Formula (3):

$$X(t,f) = \int\limits_{-\infty}^{\infty} w(t-\tau)x(\tau)e^{-j2\pi f\tau}d\tau \tag{3}$$

where $X(t,f)$ is the Fourier transform of $w(t-\tau)x(\tau)$. The formula for calculating the spectrum is:

$$SP_x(t,f) = |X(t,f)|^2 \tag{4}$$

The formula for calculating the PSD of the EEG signal is:

$$p = \int\limits_{-\infty}^{\infty} S(w)dw = \frac{1}{T}\int\limits_{-\tau}^{\tau} X^2(t)dt \tag{5}$$

where $X^2(t)$ is calculated by Formula (3) and $S(w)$ is the power spectral density of the signal. The time of motor imagery in the original signal of the 31-electrode 500 Hz sampling rate was intercepted into 3 s sample segments, and the Pwelch function in python was used to find the power spectral density for each epoch of each electrode. Considering the characteristics that the larger the window width, the higher the frequency domain resolution and the more segments are divided, the smaller the noise, we selected a hamming window with a length of 500 sampling points.

This study compared the energy values of the motor imagery phase (3 s) in various levels of abstraction of visual guidance MI tasks using frequency domain analysis. The study concentrated first on the spatial features of the 4 frequency bands delta (1–3 Hz), theta (4–7 Hz), alpha (8–13 Hz), and beta (14–25 Hz) and then on the mental load related to theta waves in the frontal region as well as theta and alpha waves in the temporal lobe, which were studied to provide a clearer picture of the differences between the different mental load states in each frequency band.

*3.7. Classification and Data Analysis*

3.7.1. SVM Classification

We classified EEG signals using SVM and performed k-fold cross validation to gauge the effectiveness of the classification. We input the ERD values into the SVM classifier. SVM is suitable for small sample data, such as EEG, and only needs a few training examples to produce accurate classification results. The radial basis function (RBF) kernel was selected as the kernel function of SVM, where the selection of the best penalty factor c and the best kernel parameter g is determined by grid optimization each time the SVM model is trained. K-fold cross-validation reduces variance by averaging the training results of k different groups, so the model's performance is less sensitive to the data division. When we divided EEG data into training and test sets, we adopted the k-fold cross-validation method where k equaled 5. Therefore, we divided the EEG data into 5 equal parts (29 trials), one of which we selected as the test set each time; we used the remaining 4 (115 trials) as the model training set. The average of the five groups of test results was taken as the accuracy of the test set. Finally, statistical studies were performed to determine the impact of various levels of abstraction of visual guidance on the ERD features derived from motor imagery on the classification of left- and right-handed MI.

3.7.2. Statistical Analysis

The experimental data were analyzed using traditional statistical methods and were performed using SPSS software. The ERD features were analyzed using the Wilcoxon signed-ranks test, since the Wilcoxon signed-ranks test is suitable for analyzing a small sample of data. The results of EEG for mental load features were analyzed using repeated-measures ANOVA. When a significant overall effect was observed, the Tukey post hoc

test was applied to examine the multiple comparisons of means. The classification accuracy was analyzed using paired-samples *t*-tests. We may determine whether there is a significant difference between the means of the two groups using the *t*-test, which uses the t-distribution theory to introduce the probability of occurrence of the difference. We adopted $p < 0.05$ as statistically significant and $p < 0.001$ as very significant. We report exact $p$ values derived from statistical tests, except when they were minuscule. To further analyze the relationships between EEG features and classification accuracy, we used the Pearson test in the analysis since the Pearson test can be used to evaluate whether two sets of data conform to a linear relationship.

## 4. Results

### 4.1. ERD Feature Extraction and Analysis

The ERD patterns of the subjects during the experiment were measured and analyzed. The EEG recorded from the C3 electrode when envisioning a right-handed grip and the EEG recorded from the C4 electrode when imagining a left-handed grasp were chosen for study based on the spatial specificity of the motor imagery. The mean ERD values of the alpha band for each of the three visual guidance groups were computed using 144 EEG data for each of the 3 levels of abstraction of the visual guidance paradigm for each participant. Their descriptive statistics are displayed in Table 1 with the three paradigms.

**Table 1.** Descriptive characteristics of ERD values for C3 and C4 channels under three levels of abstraction of visual guidance.

| Channel | Level of Abstraction | M $\pm$ SD (*N* = 17) | Median |
|---|---|---|---|
| C3 | High | 0.8732 $\pm$ 0.5671 | 0.8990 |
| | Medium | 0.9563 $\pm$ 0.7157 | 0.7897 |
| | Low | 1.3026 $\pm$ 1.1994 | 1.0092 |
| C4 | High | 0.9799 $\pm$ 1.0466 | 0.7855 |
| | Medium | 1.1164 $\pm$ 0.9910 | 0.8905 |
| | Low | 1.2976 $\pm$ 1.1627 | 1.1872 |

Figure 3 displays the ERD patterns produced by the C3, C4, and Cz channels during left- and right-hand MI under the 3 types of visual guidance in a representative 20-year-old subject to demonstrate this result. The orange line denotes the left-hand MI, whereas the blue line denotes the right-hand MI.

A Wilcoxon signed-rank test was conducted on the ERD values of the C3 channel when imagining a right-handed grasp and the ERD values of the C4 channel when imagining a left-handed grasp under three types of visual guidance paradigms for the level of abstraction (high, medium, and low). When visualizing a right-handed grasp, the median ERD value for the C3 channel was 0.8990 for the high level of abstraction group, 0.7897 for the medium level of abstraction group, and 1.0092 for the low level of abstraction group; the mean ERD value for the C3 channel was 0.8732 for the high level of abstraction group, 0.9563 for the medium level of abstraction group, and 1.3026 for the low level of abstraction group. The Wilcoxon signed-rank test revealed no differences between the high and medium level of abstraction groups ($Z = -6.86$, $p = 0.492$) and no differences between the medium and low level of abstraction groups ($Z = -1.349$, $p = 0.177$) but a significant difference between the high and low level of abstraction groups ($Z = -1.965$, $p = 0.049 < 0.05$). When imagining left-handed grasping, the median ERD value for the C4 channel was 0.7855 for the high level of abstraction group, 0.8905 for the medium level of abstraction group, and 1.1872 for the low level of abstraction group; the mean ERD value for the C4 channel was 0.9799 for the high level of abstraction group, 1.1164 for the medium level of abstraction group, and 1.2976 for the low level of abstraction group. Using the Wilcoxon signed-rank test, there was no significant difference between the high level and the medium level of abstraction groups ($Z = -0.828$, $p = 0.407$) and no significant difference between the medium and the low level of abstraction groups ($Z = -0.497$, $p = 0.619$) but a

significant difference between the high and the low level of abstraction groups ($Z = -2.012$, $p = 0.044 < 0.05$)

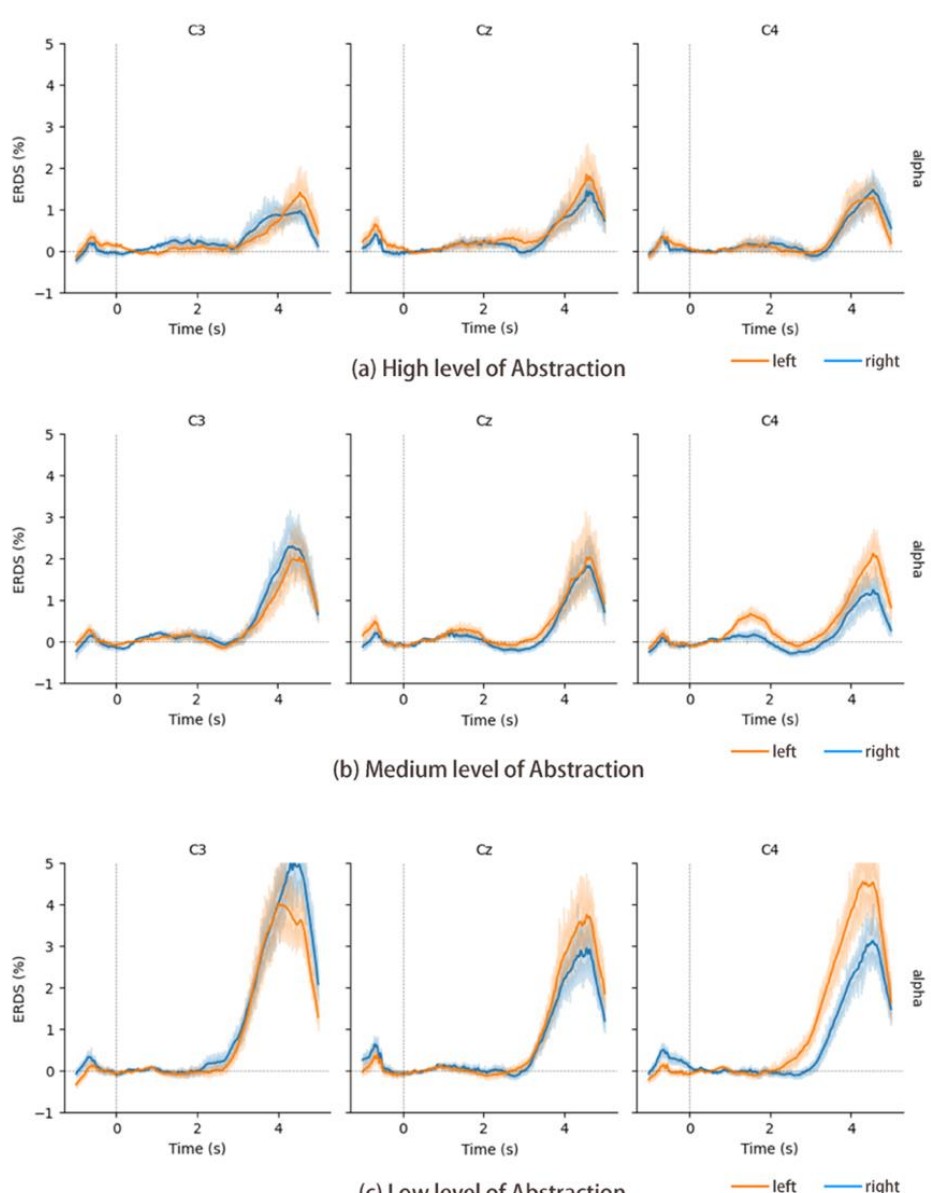

**Figure 3.** Three visual guidance ERD patterns in the brain.

### 4.2. Mental Load Features and Analysis

PSD represents the distribution of power as a function of frequency, and it is the most commonly used feature [50]. The subject's brain's overall PSD during motor imagery was estimated after analyzing the frequency domain characteristics. Figure 4 shows that between 7.5 Hz and 12.5 Hz, the subjects' brain energy peaked, and the entire brain was active. The outcome extracts a PSD for each band from the four frequency bands including theta, delta, alpha, and beta. Since the power changes of the alpha band (decreased) and theta band (increased) may serve as the discriminant indicators for workload estimation with increasing workload, we paid more attention to the power of these two bands in our results [49].

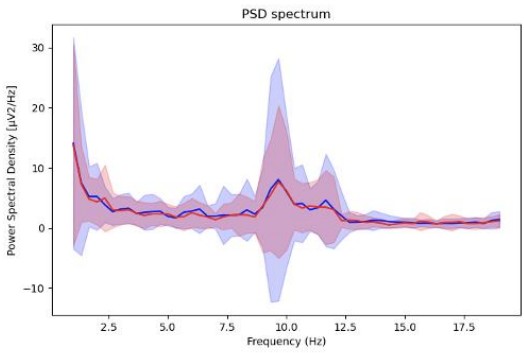

(a) High level of abstraction Group

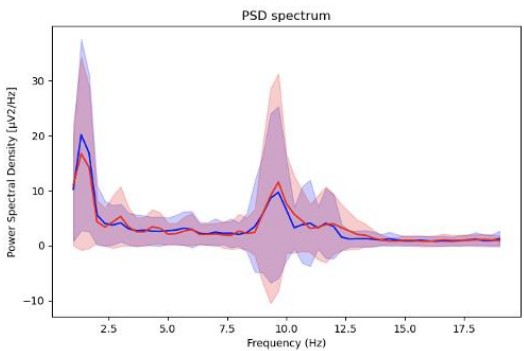

(b) Medium level of abstraction Group

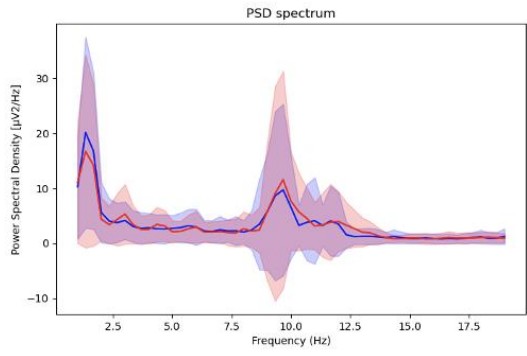

(c) Low level of abstraction Group

**Figure 4.** PSD values of the brain under three visual guidances. The red line and red shadow represents the left-hand MI, and the blue line and blue shadow represents the right-hand MI. The shadow interval represents the standard deviation. Delta waves (0.5–4 Hz) vibrate at low frequencies but high amplitudes. Therefore, there are high PSD values in the frequency range <2.5 Hz.

### 4.2.1. Theta Band Energy Analysis

Previous studies indicate that the frontal and parietal areas are sensitive to workload change [51]. As such, we used the related channels in these areas for mental load analysis. It has been hypothesized that more theta waves take place in the brain's frontal regions when people are under a heavy mental load. Therefore, we isolated eight electrode channels from the frontal area: F7, F8, F3, F4, FC5, FC1, FC2, FC6, and computed the mean PSD during the motor imagery (3–6 s of the single trial) time window. Table 2 displays the descriptive statistics for these electrode channels' mean PSD. The mean PSD during motor imagery was calculated for three levels of abstraction of visual guidance (high, medium, low) in a repeated-measures ANOVA for 8 EEG channels (F7, F8, F3, F4, FC5, FC1, FC2, FC6). The independent variables were the abstraction level of the visual guidance and the EEG channels. The dependent variable was the mean theta wave energy of each subject.

**Table 2.** Descriptive characteristics of average PSD of 8 frontal lobe channels under three levels of abstraction of visual guidance ($\mu V^2 / Hz \times 10^9$).

| Channel | Level of Abstraction | M ± SD (N = 17) |
|---|---|---|
| F3 | High | 63.28 ± 76.51 |
| | Medium | 67.71 ± 96.97 |
| | Low | 52.64 ± 35.81 |
| F4 | High | 66.55 ± 77.96 |
| | Medium | 73.64 ± 106.11 |
| | Low | 55.49 ± 36.02 |
| F7 | High | 53.86 ± 69.95 |
| | Medium | 53.45 ± 79.77 |
| | Low | 41.50 ± 34.23 |
| F8 | High | 50.05 ± 55.37 |
| | Medium | 50.34 ± 67.23 |
| | Low | 40.51 ± 27.19 |
| FC1 | High | 66.38 ± 77.74 |
| | Medium | 73.21 ± 98.11 |
| | Low | 54.96 ± 34.74 |
| FC2 | High | 66.38 ± 77.74 |
| | Medium | 75.80 ± 104.67 |
| | Low | 54.74 ± 34.94 |
| FC5 | High | 46.42 ± 64.64 |
| | Medium | 48.10 ± 73.28 |
| | Low | 35.54 ± 34.94 |
| FC6 | High | 44.07 ± 55.94 |
| | Medium | 48.23 ± 69.58 |
| | Low | 34.78 ± 23.41 |

The findings demonstrated that the level of abstraction of visual guidance had a significant main effect, $F = 7.228$, $p = 0.008 < 0.05$. The main effect of the electrode channel was not significant, $F = 0.447$, $p = 0.850$, and the interaction effect between the level of abstraction of visual guidance and the electrode channel was not significant, $F = 0.008$, $p = 1$. The results are also shown in Figure 5 below.

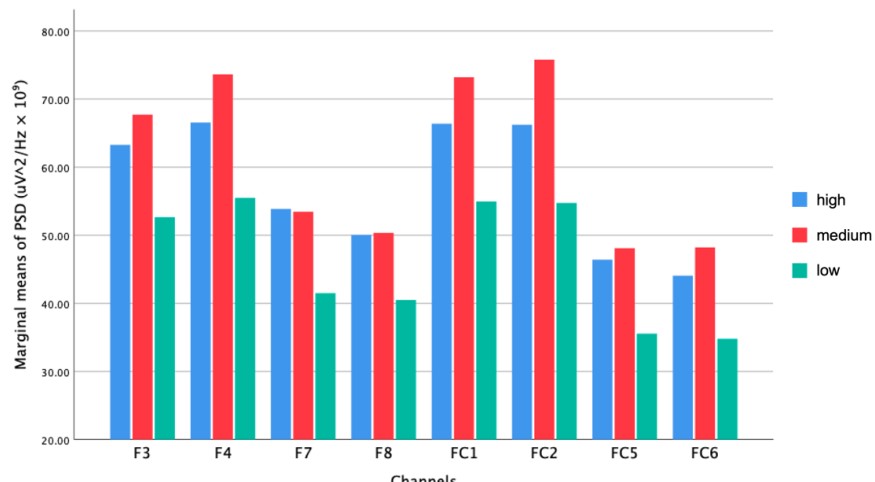

**Figure 5.** Marginal means of PSD for 8 frontal area channels under three types of visual guidance.

### 4.2.2. Theta/Alpha Band Energy Ratio Analysis

Ke et al. [52] discovered considerable variations in the theta and alpha band energies for various task kinds and levels of difficulty. To calculate the ratio of the average PSD of the theta wave to the average PSD of the alpha wave during the time window of motor imagery (3–6 s of the single trial), we extracted seven electrode channels distributed in the central region: C3, C4, Cz, CP1, CP2, CP5, CP6. Their descriptive statistics are shown in Table 3.

On three levels of abstraction of visual guidance, the ratio of the theta wave mean PSD to the alpha wave mean PSD during motor imagery was calculated (high, medium, low) with repeated measurements for the seven EEG channels (C3, C4, Cz, CP1, CP2, CP5, CP6). The independent variables were the abstraction levels of the visual guidance and the EEG channels. The dependent variables was each subject's mean theta/alpha band energy ratio.

**Table 3.** Descriptive characteristics of theta/alpha PSD ratios for the seven central zone channels under three levels of abstraction of visual guidance.

| Channel | Level of Abstraction | M ± SD (*N* = 17) |
| --- | --- | --- |
| C3 | High | 0.4928 ± 0.3311 |
| | Medium | 0.4674 ± 0.3432 |
| | Low | 0.4198 ± 0.3505 |
| C4 | High | 0.4846 ± 0.3054 |
| | Medium | 0.4591 ± 0.3048 |
| | Low | 0.4125 ± 0.3170 |
| CP1 | High | 0.4705 ± 0.3048 |
| | Medium | 0.4351 ± 0.3013 |
| | Low | 0.3876 ± 0.3182 |
| CP2 | High | 0.4534 ± 0.2851 |
| | Medium | 0.4227 ± 0.2824 |
| | Low | 0.3709 ± 0.2922 |
| CP5 | High | 0.5007 ± 0.3118 |
| | Medium | 0.4626 ± 0.2965 |
| | Low | 0.4187 ± 0.3158 |
| CP6 | High | 0.4841 ± 0.2970 |
| | Medium | 0.4526 ± 0.2809 |
| | Low | 0.4098 ± 0.3026 |
| Cz | High | 0.5360 ± 0.3674 |
| | Medium | 0.4987 ± 0.3717 |
| | Low | 0.4393 ± 0.3568 |

The level of visually guided abstraction caused significant differences in the subjects' mental load during motor imagery, $F = 44.906$, $p < 0.001$, as shown in Figure 6, with the mean energy ratio in the low abstraction group being significantly lower than in the other two levels. The main effect of the electrode channel was not significant, $F = 0.100$, $p = 0.996$, and the interaction between the electrode channel and the level of abstraction was not significant.

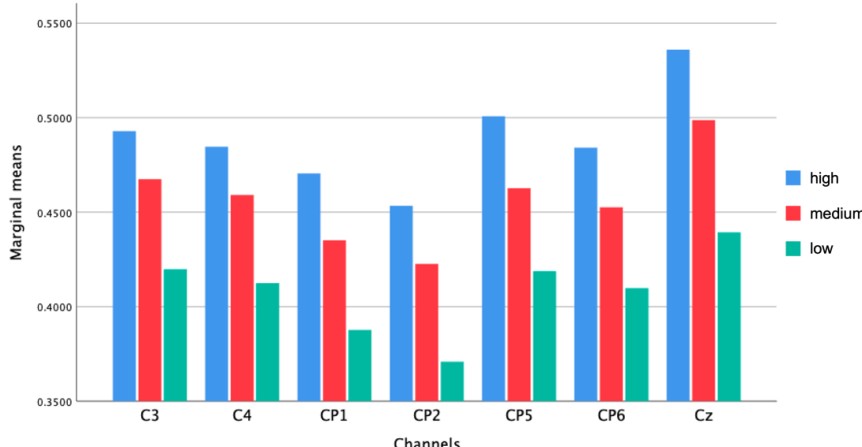

**Figure 6.** Marginal averages of Theta/Alpha wave energy ratios for the central zone channel under three types of visual guidance.

*4.3. Classification Performance*

Table 4 displays the accuracy of motor imagery categorization under the visual guidance of individuals with various levels of abstraction. The highest classification accuracy of the group with a high level of visual guidance abstraction was 90.7143%, and the average accuracy was 85.2381%; the highest classification accuracy of the medium-level visual guidance group reached 90.7143%, and the average accuracy was 86.5294%; the highest classification accuracy rate of the low-level visual guidance group was 97.1429%, and the average accuracy was 90.5732%. Figure 7 shows the analysis of the classification model by the ROC curve. Figure 8 demonstrates that the low-level abstraction of the visual guidance group's experimental findings had a greater average categorization accuracy than the other two levels of abstraction of visual guidance. The classification accuracy of motor imagery under the three levels of abstraction of visual-guided paradigms was tested pairwise using paired-samples *t*-tests (high, medium, and low).

**Table 4.** Descriptive statistics of the accuracy of MI classification under three levels of abstraction of visual guidance.

| Level of Abstraction | M $\pm$ SD (N = 17) | Min | Max |
|:---:|:---:|:---:|:---:|
| High | 0.8524 $\pm$ 0.0372 | 0.7857 | 0.9071 |
| Medium | 0.8653 $\pm$ 0.0386 | 0.7714 | 0.9071 |
| Low | 0.9057 $\pm$ 0.0495 | 0.7786 | 0.9714 |

The findings revealed that there was no significant difference between the high and medium levels of visual guidance abstraction groups ($p = 0.216$); there was a significant difference between the low level and the high level of visual guidance abstraction groups ($p < 0.001$); and there was a significant difference between the medium level and the low level of visual guidance abstraction groups ($p = 0.004 < 0.05$).

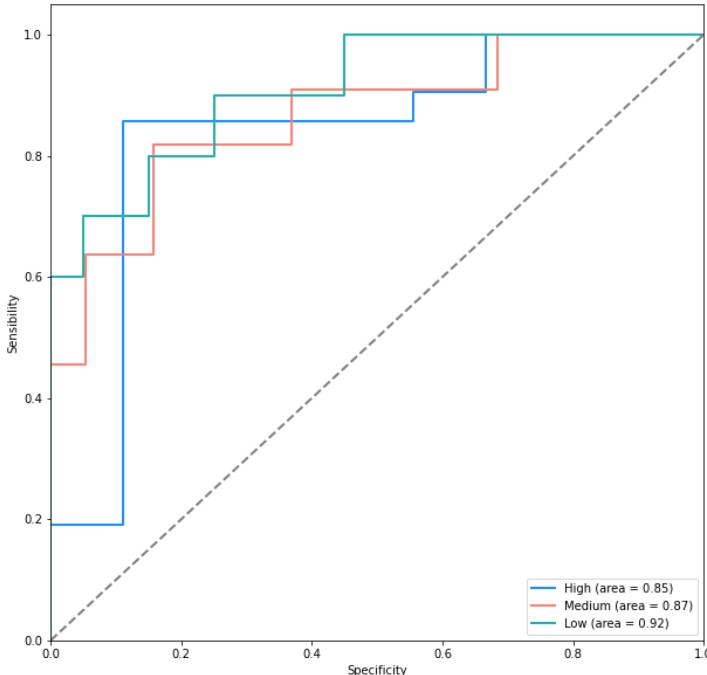

**Figure 7.** ROC curves for classification of the test set under three levels of visual guidance.

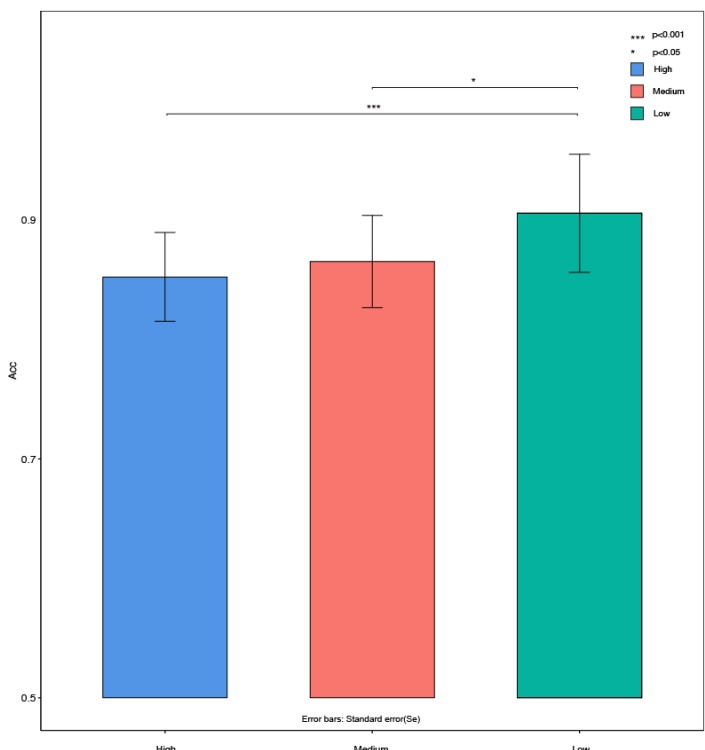

**Figure 8.** Average ACC under three levels of visual guidance.

### 4.4. Correlation between Brain Activity, Mental Load, and MI-BCI Performance

We explored the relationships between the ERD values in the central area of the brain, which represents brain activity, and the MI-BCI performance and the two measurements of mental load. Pearson's correlation coefficient for the investigation of the relationship between the ERD values in the central area of the brain and the accuracy rate was found to be $r = 0.249$, $p = 0.012 < 0.05$. This indicates that the ERD values in the central area of the brain and the accuracy rate have a positive correlation (see Figure 9). More brain activity improves the classification accuracy and functionality of MI-BCI.

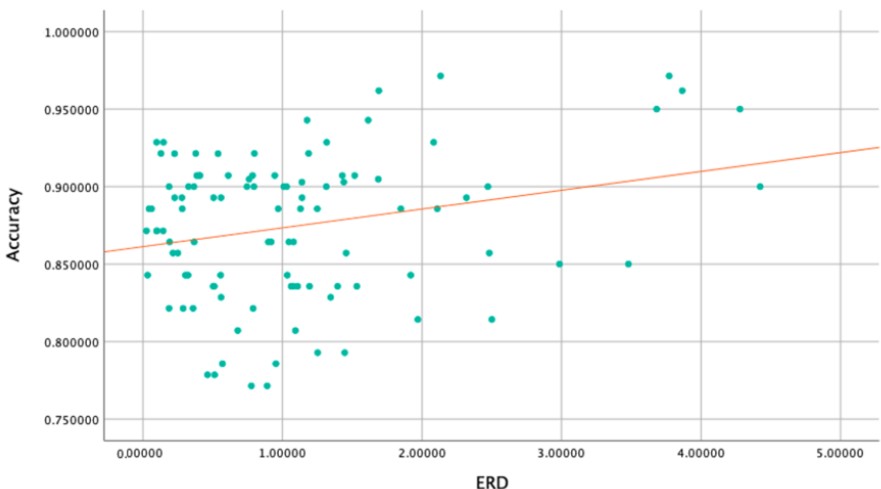

**Figure 9.** Correlation between ERD and accuracy.

In the correlation analysis between the PSD of the frontal theta wave and the accuracy rate, the Pearson correlation coefficient of the obtained results was $r = -0.109$, $p = 0.007 < 0.05$. It is inferred from this that the PSD of the frontal theta wave has a negative correlation with the accuracy rate. In the correlation analysis between the ratio of the

PSD of the theta wave and alpha wave in the central area and the accuracy, the Pearson correlation coefficient of the obtained result was $r = -0.278$, $p < 0.001$. It is inferred from this (shown in Figures 10 and 11) that the ratio of the PSD of the theta wave and alpha wave negatively correlates with the accuracy. The lower the ratio of the PSD of the theta wave to the alpha wave in the central area, the higher the classification accuracy and the better the performance of MI-BCI.

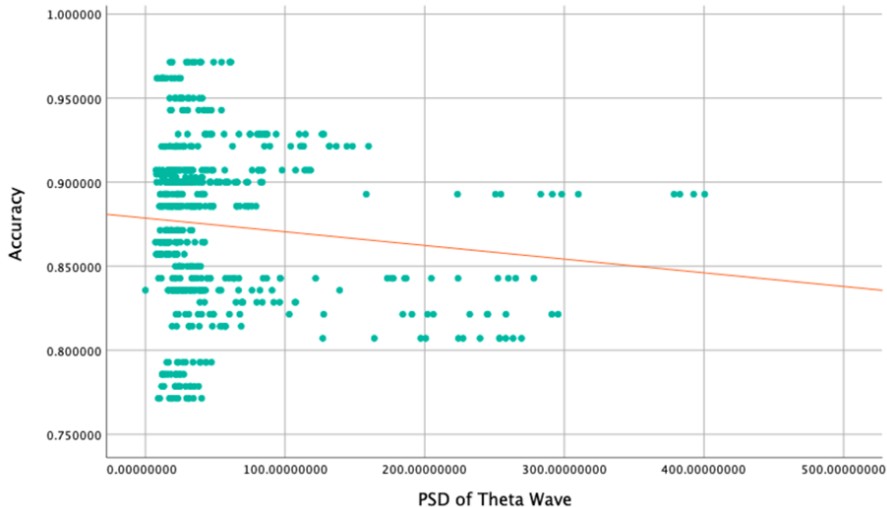

**Figure 10.** Correlation between theta wave PSD and accuracy.

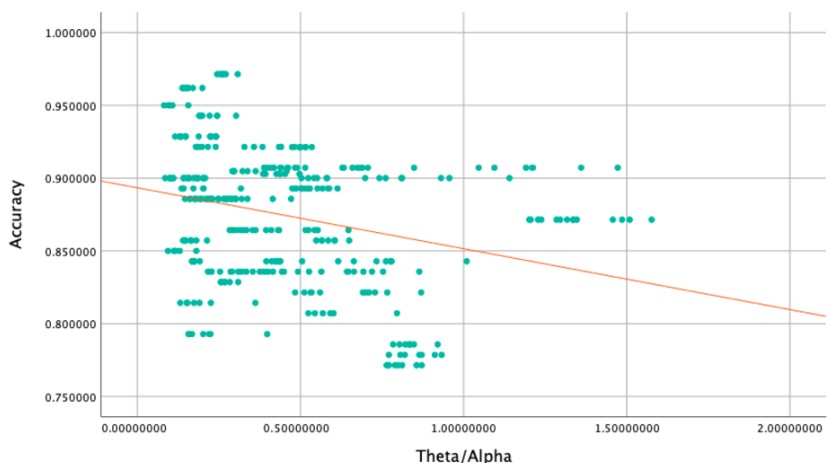

**Figure 11.** Correlation of Theta/Alpha ratio with Accuracy.

## 5. Discussion

### 5.1. Brain Activity

During the experiment, the ERD patterns of the participants were measured and analyzed to evaluate hypothesis H1 that brain activity is connected with the level of abstraction of visual guidance. The generated graphs resemble each other. The temporal performance of ERD activation in the brain was comparable throughout the three experimental settings, although ERD amplitudes varied (shown in Figure 3). When visual guidance with a high level of abstraction was used, the amplitude of ERD was relatively low. The amplitude of ERD increased as the level of abstraction decreased, indicating an increase in brain activation. Therefore, visual guidance with varying levels of abstraction impacted the subjects' brain activity during the MI task.

### 5.2. Mental Load

Two metrics were utilized in the analysis to evaluate H2, which states that users' cognitive load is connected to their level of abstraction of visual guidance. The PSD of each frequency band and channel were extracted using frequency domain analysis. Through statistical analysis, it was discovered that the average mental load of the group with a low abstraction level differed significantly from the mental load of the group with a high abstraction level. In contrast to the other two groups, the subjects in the group with a high level of abstraction had to switch the primary body throughout the cognitive process to translate the ball's movement into the hand's movement. Such a mental process will almost certainly result in some mental burden.

### 5.3. MI-BCI Performance

This study used classification accuracy, a conventional BCI performance metric, to test H3 that MI-BCI performance is connected with the level of visual-guided abstraction. The experiment's findings indicate that the level of visual guidance abstraction impacted MI-BCI performance: the groups with low and medium levels performed better than groups with a high level of visual guidance abstraction. Regarding average accuracy, the low abstraction group's classification accuracy was about 5% higher than the high abstraction group's, and the medium abstraction group's classification accuracy was around 1% higher than the high abstraction group's. Therefore, on the motor imagery task, the performance of MI-BCI is influenced by visual guidance with different levels of abstraction. The experimental paradigm for asynchronous motor imagery was used in this study. We focused more on the time of motor imagery while calculating and evaluating the BCI's MI performance, and a blank screen was employed to reduce the visual effect of visual guidance and motor imagery-related brain function. Since there was no discernible difference between the groups with medium and low levels of visual-guided abstraction, additional research utilizing a synchronized motor imagery paradigm and other metrics for assessing MI-BCI performance may shed light on the visual-guided pairing and the potential effects of BCI performance.

### 5.4. Correlations

To study H4 and validate earlier related research, which found that the performance of the MI-BCI is negatively impacted when subjects are under a heavy mental load [30], we attempted to explore the correlation results by drawing the brain electromagnetic topographies during the MI period to make a discussion. The statistical analyses in the Results section revealed a specific correlation between the subjects' ERD values in the central area of the brain and the classifiability of their motor imagery. The classifiability of motor imagery is negatively correlated with the subjects' mental load, as demonstrated by the following: the mental load in the frontal lobe area of the brain and the mental load in the central area of the brain are both negatively correlated with the classifiability of motor imagery, and related research proposed that attention is closely tied to the frontal lobe region of the brain [51]. We found during the early MI period (2.0–2.9 s), the frontal lobe region of the brain was highly active (shown in Figure 12). The higher the level of abstraction, the greater the amplitude of theta waves in the frontal regions and the deeper red on the brain's electromagnetic topographies. The results presented in the brain electromagnetic topographies echo those in the Results section. The data suggest that to complete the visually guided MI task successfully, subjects must maintain high concentration; high mental load results in lower classification accuracy.

The central region of the brain contains human sensorimotor-related regions, and MI tasks cause the central region to become active. As shown in Figure 13, in the middle and late period of motor imagery (3.5–5 s), the brain's alpha wave showed high activity. As the level of abstraction decreases, the activity of alpha waves in the central area of the brain increases. Based on the findings mentioned in the Results section, it is hypothesized that

participants will be able to complete MI activities and exercise more easily the less mental load there is in the central area of the brain. Additionally, the imagery is more classifiable.

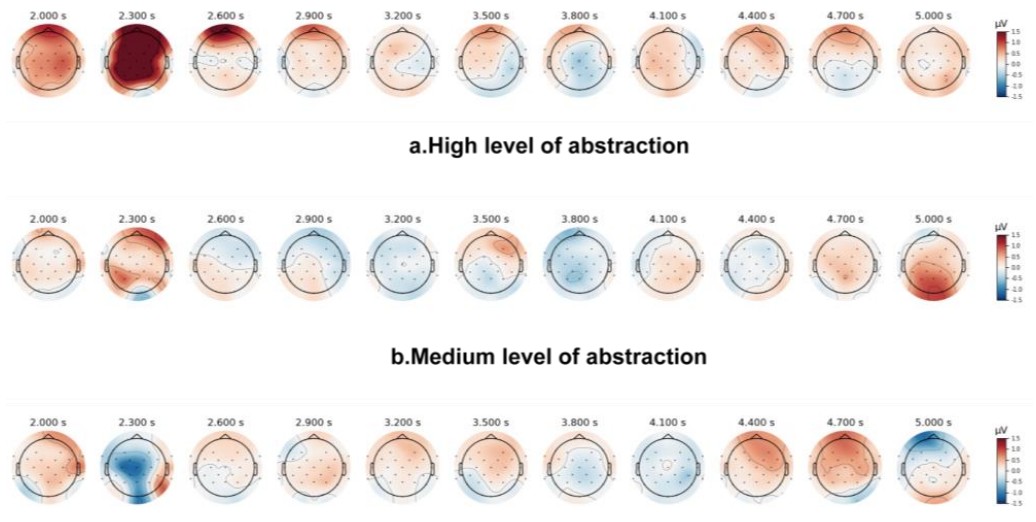

**Figure 12.** The brain electromagnetic topographies of the theta band of the three visual guidance groups.

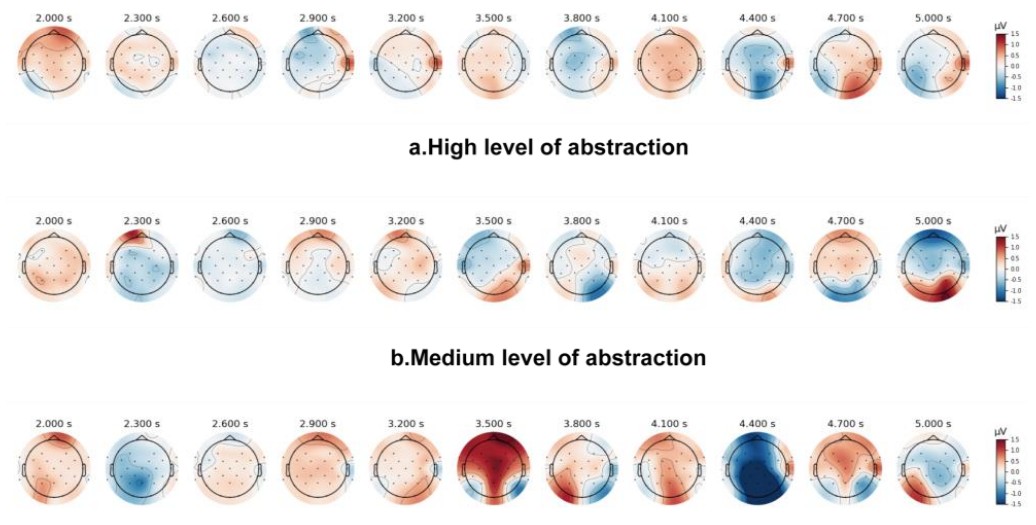

**Figure 13.** The brain electromagnetic topographies of the alpha band of the three visual guidance groups.

### 5.5. Research Limitations and Future Work

Future research can investigate additional metrics that have not been examined, even if this study offers preliminary insights into the level of visual-guided abstraction on the interaction between the two subsystems, machine, and user in an asynchronous MI-BCI system. Calculating the continuous mental load may be more appropriate for actual use situations. This study used 17 healthy volunteers because according to related research, the brain mechanisms used by healthy persons and those with disabilities when performing motor imagery are the same. Due to the complexity of this experiment and the long time required by the three tasks, data were collected from only a small sample (17 subjects), which led to the abnormal distribution of ERD data. If more volunteers and individuals

with disabilities participated, other statistical analysis methods with post hoc test might be more accurate because the brain–computer interface based on motor imagery is more applicable to people with disabilities.

A series of methods can be further explored in the future including modeling the abstraction degree of visual guidance suitable for subjects according to indicators such as brain activity and mental load to realize the function of recommending more suitable guidance for subjects. As for modeling methods, these can be weakly supervised deep learning-based classification methods, other nonparametric heuristics, or experience-based methods in future work.

## 6. Conclusions

This study examined how an MI-BCI system was affected by the visual-guided system's level of abstraction (user brain activity, mental load, BCI performance). This study aimed to investigate the processes of users during motor imagery, to comprehend the changes in user brain activity (ERD), brain load, and the performance of the overall BCI system induced by visual guidance of the level of abstraction, as well as to investigate user mechanisms and whether the brain activity and mental load will affect the performance of BCI. Indicators were primarily used to examine the study's objectives, including the ERD, PSD for the brain region, and classification accuracy. We developed three different levels of abstraction of visual-guided motor imagery experiments to investigate the four hypotheses we proposed, and the four hypotheses listed above were supported by the variations in indicators across several experimental settings. The results showed that during the MI task, the brain activity of the subjects was affected by visual guidance with different levels of abstraction, and visual guidance with a low level of abstraction caused more intense brain activity in the users; level of abstraction of visual guidance had an impact on the subjects' brain loads, where the visual guidance with a low level of abstraction resulted in a lower brain load for the user; level of abstraction in visual guidance had an impact on MI-BCI performance, and low-abstraction visual guidance also had better mean classification performance. The performance of MI-BCI has a specific correlation with the subjects' brain activity and mental load.

This study's findings demonstrate that the brain is active during MI tasks, pointing to the level of visually guided abstraction as a critical variable that may influence user brain activity, mental load, and MI-BCI performance. The findings indicate that visual guidance with a low level of abstraction can be employed in brain–computer interface users' training to increase the level of brain activation while simultaneously ensuring the availability of MI-BCI to give users a better user experience and reduce the information processing load of users to achieve better brain–computer interaction while using MI-BCI. Additionally, the findings may assist those who are disabled in regaining their capacity for sustainable action. Our research opens up new avenues for sustainable MI-BCIs, which has exciting implications for extending research from academia to the applied domain.

**Author Contributions:** Conceptualization, C.Y. and L.K.; methodology, L.K.; software, Z.Z.; validation, L.K. and Z.Z.; resources, C.Y.; data curation, L.K. and Z.Z.; writing—original draft preparation, L.K.; writing—review and editing, C.Y., X.C., Y.T., L.K. and Z.Z.; visualization, L.K.; project administration, C.Y.; funding acquisition, C.Y. and X.C. All authors have read and agreed to the published version of the manuscript.

**Funding:** This research was funded by the Scientific Research Foundation of Zhejiang University City College (No.X-202203).

**Institutional Review Board Statement:** The study was conducted in accordance with the Declaration of Helsinki and approved by the Institutional Review Board of Zhejiang University.

**Informed Consent Statement:** Informed consent was obtained from all subjects involved in the study.

**Data Availability Statement:** The datasets generated for this study are available on request to the corresponding author.

**Conflicts of Interest:** The authors declare no conflict of interest.

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
