# Peer review of "Exploring the Visual Guidance of Motor Imagery in Sustainable Brain–Computer Interfaces"

_sustainability, doi:10.3390/su142113844_

Round 1
Reviewer 1 Report
Yang et al., utilized a motor imagery-based brain computer interfaces in a group of healthy adults. By systematically investigating the effect of the level of abstraction of the trigger on the BCI performance, the authors find a low level of abstraction was associated with a highest brain response in the form of ERD, and lowest level of mental load in the form of PSD, and the overall accuracy rate was 97%.
I only have a few minor points for the authors to consider:
Introductions:
Lines 43-46:
“when picturing unilateral hand movements, the energy of mu rhythms (8–12 Hz) and Beta rhythms (14–30 Hz) in the contralateral brain region is lowered, whereas the energy of mu rhythms and Beta rhythms in the ipsilateral motor-sensory areas is raised”
Here, the authors may use ERD/ERS to stand for the power suppression/enhancement.
Line 105:
Should use Liang et al., rather than Shuang Liang et al. Same citation issues occurred in other parts of the manuscript.
Methods:
Line 301-302
Why use TP9 TP10 as reference channels? They should contain some brain activities.
Line 310-316
Why only use mu ERD? How about the performance when using beta ERD?
Line 324-344
It is still not clear why PSD of frontal channels can be a measure of mental load, which period of brain activity was extracted for PSD calculations?
Line 370
Delta should be 1-3Hz? As the authors have use high pass filter at 0.5 Hz, 0.1 Hz does not make sense.
Results:
Have authors performed spatial filter such as Laplacian filter before using C3/C4 data in the analysis?
Table 1:
ERD should be a negative value (A-R/R), but here all data were positive.
Have authors use any assessments to evaluate the level of vividness of motor imagery? It should be a potential cofound for BCI performance.
Author Response
We sincerely appreciate the reviewer for your thoughtful comments, efforts, and time. We respond to your questions and concerns one-by-one in the file below.

Reviewer 2 Report
This is an interesting study, however the authors should make some clarifications:
1. In line 247 what was the rationale for choosing the different levels of abstraction? Please include it in the reference.
2. In fig 3 pls clarify the reason for high PSD in the frequency range < 2.5 Hz for all 3 groups.
3. Pls include the denoising method used here.
4. Pls include the t test significance in the fig.6
5. It would be appropriate to include weakly supervised deep learning pipelines to better classify the abstractions.
Author Response

(The authors gave the same response as above.)

Reviewer 3 Report
see attached file, please

Author Response

(The authors gave the same response as above.)

Round 2
Reviewer 3 Report
see the attached file, please

Author Response

(The authors gave the same response as above.)

Round 3
Reviewer 3 Report
1. Please, check the line 423-424 "The classification accuracy Paired samples t-tests." It seems as no verb is in the sentence.
2. Please, add the work's limitations in the discussion section.
Author Response

(The authors gave the same response as above.)
